# Novel Mutations in MPT64 Secretory Protein of *Mycobacterium tuberculosis* Complex

**DOI:** 10.3390/ijerph20032530

**Published:** 2023-01-31

**Authors:** Noor Muhammad, Muhammad Tahir Khan, Sajid Ali, Taj Ali Khan, Anwar Sheed Khan, Nadeem Ullah, Hassan Higazi, Sara Ali, Salma Mohamed, Muhammad Qasim

**Affiliations:** 1Department of Microbiology, Kohat University of Science and Technology, Kohat 26000, Pakistan; 2Zhongjing Research and Industrialization Institute of Chinese Medicine, Zhongguancun Scientific Park, Meixi, Nanyang 473006, China; 3Institute of Molecular Biology and Biotechnology (IMBB), The University of Lahore, KM Defense Road, Lahore 54770, Pakistan; 4Department of Microbiology & Biotechnology, Bacha Khan University, Charsadda 24550, Pakistan; 5Institute of Basic Medical Sciences, Khyber Medical University, Peshawar 25100, Pakistan; 6Department of Clinical Microbiology, Umeå University, 90185 Umeå, Sweden; 7Department of Medical Laboratory Sciences, College of Health Sciences, Gulf Medical University, Ajman P.O. Box 4184, United Arab Emirates

**Keywords:** MPT64, mutations, diagnosis, genomes, mycobacterium, tuberculosis

## Abstract

Tuberculosis (TB) is a global health problem caused by the *Mycobacterium tuberculosis* complex (MTBC). These bacteria secrete various proteins involved in the pathogenesis and persistence of MTBC. Among the secretory proteins, MPT64 (Rv1980C) is highly conserved and is also known as a major culture filtrate that is used in rapid diagnosis of MTBC. In the current study, we aimed to find the mutation in this highly conserved protein in isolates from the Pashtun-dominant province of Pakistan. We analyzed 470 *M. tuberculosis* whole-genome sequences of Khyber Pakhtunkhwa Province. Mutations in the MPT64 gene were screened through TB-Profiler and BioEdit software tools. The DynaMut web server was used to analyze the impact of the mutation on protein dynamics and stability. Among 470 MTB genomes, three non-synonymous mutations were detected in nine isolates, and one synonymous mutation (G208A) was found in four isolates. Mutation G211T (F159L), which was detected at the C-terminal domain of the protein in six isolates, was the most prominent. The second novel mutation, T480C (I70V), was detected in two isolates at the C-terminal side of the protein structure. The third novel mutation, A491C (L66R), was detected in a single isolate at the N-terminal side of the MPT64 protein. The effect of these three mutations was destabilizing on the protein structure. The molecular flexibility of the first two mutations increased, and the last one decreased. MPT64 is a highly conserved secretory protein, harboring only a few mutations. This study provides useful information for better managing the diagnosis of MTB isolates in high TB-burden countries.

## 1. Introduction

Tuberculosis is a global health problem caused by *Mycobacterium tuberculosis* complex (MTBC). It targets mainly the lungs (Pulmonary TB); however, it can also affect other sites of the body (Extra-pulmonary TB) [1]. Globally, more than 10 million people have developed TB, and 1.4 million people have died.

Pakistan has 5th position among high TB burden countries in the word, and of the four provinces of Pakistan, Khyber Pakhtunkhwa (KP) is the one that contributes a portion of 11.9% of the total national population [2]. Multi-drug resistant tuberculosis (MDR-TB) is a type of TB in which the MTBC shows resistance to Isoniazid and Rifampicin. The organism that is resistant to Isoniazid, Rifampin, Fluroquinolone, and Bedaquiline or Linezolid is known as extensively drug-resistant TB (XDR-TB) [3].

In many high TB burden countries with few resources, Ziehl–Neelsen (ZN) microscopy is often used because of its low cost, but the sensitivity of the test is too low. Liquid culture systems are recommended by the World Health Organization (WHO) in resource-constrained settings. For the isolation of MTBC, low-income countries still depend on Lowenstein–Jensen (LJ)medium. A new molecular assay known as Xpert MTB/RIF, which provides simultaneous detection of MTB and Rifampicin resistance, is also endorsed by the WHO [4].

Drug susceptibility testing (DST) is used to ascertain drug resistance and treatment of MDR and XDR-TB. A new molecular diagnostic tool was endorsed by the WHO in 2008, GenoType^®^ MTBDRplus (Hain Life science, Nehren, Germany), to identify MDR-TB. The second line (GenoType^®^ MTBDRsl) assay was endorsed by the WHO in 2016 and is used to detect the resistance of MTBC to second-line fluoroquinolones (FQ) and injectable drugs to diagnose the XDR-TB [5].

Similar to many other bacteria, *M. tuberculosis* has well-regulated secretion systems that are important for its virulence and pathogenesis. Recent developments in technologies have facilitated the precise detection of secretory proteins in culture filtrate [6]. MPT64 is deleted from nearly all strains of *Mycobacterium bovis* bacilli that are used for making the Bacille Calmette–Guerin (BCG) vaccine and is only expressed and secreted from actively growing cells. Loss of this region has been correlated with a drop in virulency [7].

Diagnosis of TB in resource-limited countries causes a delay in TB management and treatment, GeneXpert, culture, MTBDRplus, and MTBDRsl assays are used for the diagnosis of TB recommended by the WHO. The differentiation of MTBC from *Mycobacterium* other than tuberculosis (MOTT) is necessary even after a positive culture. Identification of MTBC from the positive culture is possible by TB-Neo assay (Tauns Laboratories, Int, Numazu, Japan) and SD Bioline assay (Standard Diagnostic, Republic of Korea). This rapid immuno-chromatographic technique (ICT) is based on MPT64 secretory protein tests for differentiating MTBC from MOTT. MPT64 is a highly conserved region of MTBC and has been reported in some previous studies. Occasionally, discrepant results are given due to mutations in the MPT64 gene, even the MTBC will give positive result by microscopy, culture, and GeneXpert techniques. This study was conducted to find the discrepant results and confirm them by MTBDRplus assay and DNA sequencing techniques. This analysis may reveal the discrepancies of ICT results with other diagnostic techniques. Such type of analysis should be performed to differentiate nontuberculous mycobacteria(NTM) from MTBC. Such discordant cases can act as chronic carriers of TB and can transmit this lethal disease [8].

In the current study, we aimed to investigate the mutation in the MPT64 gene of MTBC and its effects on the MPT64 protein of the isolates from Khyber Pakhtunkhwa (K.P.), Pakistan. The study will provide useful information about the variation in the MPT64 protein for better management of TB diagnosis in high-burden countries, including Pakistan.

## 2. Materials and Methods

In the current study, we analyzed 470 MTBC isolates, among which 130 were sequenced from KP, Pakistan, and the remaining samples were retrieved from the National Center for Biotechnology Information (NCBI) (https://www.ncbi.nlm.nih.gov/) on 10 November 2020.

A total of 8822 positive isolates during 2018 to 2020 were processed in PTRL at Hayatabad Medical Complex, Peshawar, Pakistan. Biological samples were received from all the drug-resistant centers of the province. N-acetyle-L-cystein-sodium hydroxide was used for sample digestion. LJ solid culture and liquid MGIT tubes containing 7H9 media were used to culture MTBC [9]. The isolates were tested by an ICT by the SD Bioline MPT64 assay. All the positive isolates were subjected to DST, which was performed through an automated BACTEC MGIT 960 system (BD Diagnostic, Franklin Lakes, NJ, USA) [10]. For the confirmation of the drug resistance profile of MTBC, DST was repeated, and the drugs were assessed through a well-established WHO critical concentration.

Genomic DNA was extracted by the cetyl trimethyl ammonium bromide (CTAB) method [11]. The DNA underwent whole-genome sequencing at the London School of Hygiene and Tropical Medicine (LSHTM) using the Illumina MiSeq platform with a 151 bp paired-end protocol. The majority of the selected isolates were phenotypically assessed as MDR.

### 2.1. Retrieval of Genome from NCBI

The whole genome sequenced data of 340 MTBC isolates was retrieved from NCBI (Accession numbers ERR2510340-ERR251079670). The retrieved genomes had been submitted from the country of Pakistan, and most of the samples were from the KPK Province. The MTB genomes of Pakistani isolates were submitted to NCBI under the San Raffaele Scientific Institute (CRyPTIC) project. Oxford University researchers, in partnership with Public Health England (PHE), are leading a new worldwide collaboration called CRyPTIC to speed up diagnosis of the disease and, thus, eliminate TB as a public health problem by 2035. In partnership with the University of Leeds, Brighton and Sussex University Hospitals, NHS Trust, and PHE Birmingham, the Oxford University’s researchers initially sequenced 3651 TB genomes in CRyPTIC project. The CRyPTIC study aims to collect and analyze a further 100,000 samples from across the world, providing a database of MDR-TB that will underpin diagnosis using WGS (Figure 1).

### 2.2. Analysis of Retrieved Samples

TB-Profiler was used for all the selected 470 samples to analyze the sequencing data of MTBC. It is a well-established bioinformatics tool that uses BWA/bowtie2 and SAM tools. FASTA files were formed for the final analyses of nucleotides and amino acids. More reads can be observed about the markers in the library at the time of testing, and we can look at the up-to-date CSV file [12].

### 2.3. Mutations Analysis

The sequences of all the samples were analyzed using BioEdit. The entire extracted genes of MPT64 from all isolates were aligned against the control (H37Rv) to find out position changes among these sequences [13]. Mutations in the entire gene of MPT64 were also analyzed by BioEdit, and mutations in the nucleotide sequences were marked against the reference. The Genome-wide *Mycobacterium tuberculosis* variation (GMTV) database was screened to obtain information about the novel mutations. The literature was also searched for the identification of novel mutations in the gene [14]. The reference protein of MPT64 was downloaded from NCBI. Extracted genes of MPT64 were translated into amino acids and were aligned against the reference protein of MPT64 (H37Rv). BioEdit extracted an amino acid numerical summary, and the list was analyzed for amino acid substitutions.

### 2.4. MPT64 Structure Analysis

The crystal structure of MPT64 was retrieved from the protein data bank (PDB) under the accession code 2HHI [15]. The PDB distributes coordinate data, structure factor files, and NMR constraint files. In addition, it provided documentation and derived data. The coordinate data are distributed in PDB and mmCIF formats.

### 2.5. Impact of Mutation on Protein Structure

The DynaMut webserver was used to find the effects of mutations on MPT64 stability and dynamics. DynaMut produces a consensus prediction of the impact of the mutation on the stability of protein [16].

## 3. Results

Among 470 genomes, 13 isolates harbored mutations in MPT64 gene (Table 1). Three novel non-synonyms and one synonymous mutation were detected in thirteen isolates. The accession numbers of genomes harboring the non-synonymous mutations are ERR2510511, ERR2510860, ERR2510512, ERR2510255, ERR2510448, ERR3335730, ERR3335746, ERR2510249, and ERR3335711. The accession number of the genomes carrying synonymous mutations are ERR3335, ERR2510553, ERR2510554, and ERR2510557.

In the below data (Table 2), six patients were male, and three were female; most of them were aged 15–46 years old. Mutation G211T (F159L) was detected at the C-terminal of the protein in six isolates. Out of six isolates, four were drug-resistant cases that consisted of three MDR strains registered in Cat-II. In the case of the second novel mutation, T480C (I70V), which was detected in two isolates, was present at the C-terminal side of the protein structure, in which one strain was MDR. The third novel mutation, A491C (L66R), was in a single isolate present at the N-terminal side of the MPT64 protein, and the strain was MDR.

All nine samples were positive on GeneXpert, culture, and line probe assays (Table 3). The data were aligned against the reference control (H37Rv) for identifying and characterizing mutation in positive isolates. The mutation in the MPT64 gene was due to the deletion or insertion of nucleotides at three different positions.

Mutation G211T (F159L) was reported in the MPT64 gene in six isolates, which accounts for 66% of the total nine isolates; the guanosine was changed to a thymine nucleotide at the 211th position. The mutation was present at the C-terminal side of the protein and was found novel. In this mutation, the phenylalanine (F) nonpolar amino acid was changed into nonpolar leucine (L).

Mutation T480C (I70V) was reported in two isolates, which accounts for 22% of the total isolates. This mutation was also present at the C-terminal side of the protein structure. A mutation was found at the 481st position of the gene in which the thymine was changed into a cytosine nucleotide. This type of mutation was also not reported previously. The isoleucine non-polar amino acid was changed at the 70th position into non-polar valine.

Another mutation, A491C (L66R), was reported in a single genome, which accounts for 11% of the total, and was found at the N-terminal side of the protein structure. The mutation was found at the 491st position of the gene, in which the adenine was changed into cytosine nucleotide. This type of mutation was not reported previously. The non-polar leucine amino acid was changed into polar arginine amino acid at the 66th position of the protein.

However, the mutation G208A was found in another four isolates, but there was no mutational effect on the amino acid sequence, so it may be due to the stop codon. In this mutation, the guanosine was changed into adenine at the 208th position of the gene. This was a synonymous mutation and has no effect on protein expression (Table 4).

Mutation G211T (F159L) was detected in six isolates and was analyzed by DynaMut. The effect of this mutation on protein structure was found to be destabilizing, due to the prediction outcome ΔΔG (−1.855 kcal/mol), and entropy energy between wild-type and mutant was Δ (ΔΔS_Vib_ (0.909 kcal·mol^−1^·K^−1^)). Destabilization is the decrease of the stability of a protein, making it more vulnerable to degradative processes. Mutations that alter protein function (new-function mutations), in particular, are generally destabilizing and can reduce protein and organismal fitness. Molecular flexibility of the protein was increased due to this mutation (Figure 2).

Mutation T480C (I70V) was detected in two other isolates. The effect of this mutation on protein structure was destabilizing, and the prediction outcome was (−0.381 kcal/mol). Moreover, the entropy energy between the wild-type and the mutant (0.230 kcal·mol^−1^·K^−1^) showed the stabilization effects. The protein’s molecular flexibility after the mutation also increased (Figure 3).

Mutation A491C (L66R) was detected in only one isolate. The effect of mutation on protein structure was destabilizing, as it has a prediction outcome (0.931 kcal·mol^−1^·K^−1^), and the entropy energy between the wild-type and the mutant was (−0.931 kcal·mol^−1^·K^−1^). The molecular flexibility of the protein due to this mutation may be decreased (Figure 4).

Interestingly, the G208A mutation was reported in four isolates, but the mutation was synonymous and has no effect on the protein expression.

## 4. Discussion

Tuberculosis is a major global health issue, and Pakistan has the fifth highest number of cases worldwide. There are different diagnostic tools for the diagnosis of TB, but most of them have low sensitivity. AFB smear microscopy by ZN and FM are performed in a very short time but cannot discriminate MTBC from NTM. The need for proper identification of MTBC is urgent, and diagnostic tools must be simple, sensitive, and cost effective. Modern tools such as PCR, 16S rRNA based detection, and HPLC are the most demanding methods but are more expensive and technical. It was therefore decided to identify MTBC on the basis of secretion of proteins in the media by immune-chromatographic techniques [17].

Culture and DST for MTBC from the clinical samples play an essential role in the identification and understanding of drug-resistance patterns for the better management of TB. Liquid culture is an advanced technique that is used to identify MTBC and is WHO-recommended. Clinically, the identification and differentiation of MTBC from MOTT is very important for the proper treatment of TB. Therefore, the differentiation is performed using a very simple, cost-effective, and rapid diagnostic method, especially in resource-limited countries. The differentiation of MTBC from NTM is performed based on the secretion of protein in the media. An ICT method is used for the antigen–antibody reaction. A major culture filtrate protein, MPT64 (24 KDa), is used for the identification of MTBC. SD and BD Bioline assays are used, and anti-MPT64 mouse monoclonal antibodies are used in the capilia. The sensitivity and specificity of these assays are exquisite and give the vigorous intensity of test bands against MTBC only. Few studies have evaluated the performance of these assays with 97–100% sensitivity and 100% specificity. Some false negative results are also possible due to unique mutations in the MPT64 gene.

Our analysis revealed that the mutation in the MPT64 gene is highly conserved, and very few mutations were found in our results. Some previous studies have also reported that mutations in this protein are very rare, and it is a highly conserved region of MTBC. MOTT may be found in the form of serpentine cords and also in the cluster or single form; therefore, microscopy tests are not confirmatory. ZN microscopy is a very simple and cost-effective technique for the diagnosis of TB, but it cannot discriminate between MTBC and MOTT.

We evaluated the performance of an SD Bioline MPT64 assay for the diagnosis and differentiation of MTBC. An ICT for the MPT64 antigen can be used as an alternative of conventional diagnostic tools. The sensitivity and specificity of MPT64 antigen-based assay was 100% accurate as compared to other nonspecific diagnostic tools. Our results are similar to other studies that were conducted in different laboratories of the globe. Such recommendation is also suggested by other researchers due to the simplicity of the test, low cost, and shorter time also. These characteristics make this technique more appropriate for testing MTBC in TB diagnostic laboratories [18].

All the mutated samples were found positive by ZN, GeneXpert, and MTBDRplus assay and were confirmed by next-generation sequencing. A study conducted by Muyoyeta et al., 2013 [19] and reported 4/52 (7.69%) capilia negative isolates while these were detected MTBC by LPA. Hirano K et al., 2004 [20] also reported that 12/381 (3.15%) were negative by ICT, and the confirmation was conducted by hybridization and AccuProbe tests. These studies do not correlate with our results because all the mutations of our study do not affect the capilia. They also reported that negativity of ICT for the isolates of *M. tuberculosis* was due to the mutation in the MPT64 gene. The reported mutation was 63-bp deletions and was found at nucleotides 196 to 258, and the amino acid sequence was changed from 43 to 63. The mutation was reported at position 400 (G to A), in which the Guanosine at position 400 was changed to Adenine at position number 402. This type of mutation created a stop codon at 400 to 402 (TGA). The reported point mutation at different positions affects the capilia results and gives negative results, and this was different from our study.

Qiu et al., 2015 [21] conducted a study in which eight species had negative results by capilia. The deletion in 63-bp led to a major impact on the structure of MPT64. In the study, non-synonymous mutation was detected in four samples, and this mutation has no effect on the expression of protein structure. This study is in disagreement with our results because our results do not have effects on the diagnosis by ICT.

Zamir H et al., 2022 [22] conducted a study in Pakistan, and the liquid culture medium-based samples were confirmed using ICA, in which 193 (96.5%) were positive, whereas seven (3.5%) were negative. Out of 14 sequenced samples, 7 samples were positive, and 7 samples were negative; 13 samples were identical to the reference, and just 1 sample (ICA positive) showed a C477A point mutation (F159L). The results indicate that AgMPT64 can be considered to be a potent vaccine candidate. This study correlates with our results, but the position of the mutant nucleotide is different from our results. In our results, the guanine nucleotide was changed into thymine nucleotide at the 211st position.

Chew et al., 2017 [23] performed a similar study for mutational analysis of MPT64 gene. Among six isolates, a 63-base-pair deletion (197 to 259) resulted in the truncation of the MPT64 protein. One isolate had a two-base-pair insertion at nucleotide 347, resulting in a frame shift mutation, and another isolate had a missense mutation at nucleotide position 257. The 63-base pair deletion in MPT64 appears to be the most prevalent mutation followed by non-synonymous substitutions. These mutations affect the structural topology and, subsequently, the antibody-based recognition of the MPT64 antigen by the immunochromatographic strip. The SD Bioline TB Ag MPT64 negative isolates in this study were not genotyped.

Singh et al., 2019 [8] conducted a study on 4737 culture follow-up MDR strains to evaluate the performance of capilia. Out of 4737 isolates, 889 were detected positive by liquid culture, 3375 were negative, and the remaining 473 were found to be contaminated. In total, 60 (6.7%) isolates, out of 80 positive cultures, were detected as positive by liquid culture and ZN microscopy, but the capilia were negative. In total, of 60 positive isolates, 10 (16.7%) were detected as positive by MTBDRplus and PCR, and the remaining were NTM. Whole-genome sequencing was performed for the samples, and mutation was seen in the capilia-negative isolates. The mutation in the MPT64 gene was due to deletions or insertions of nucleotides at three different positions. This study disagrees with our results.

In our results, the mutation G211T (F159L) was found in six isolates, and has not beenot previously reported as having destabilizing effects on protein structures. In two other isolates, the mutation T480C (I70V) was found to have a destabilizing effect on protein structures and has also not been reported previously. The mutation A491C (L66R) was reported in a single isolate and has a destabilizing effect on the protein structure. It was a novel mutation in MPT64 and has not been reported previously. Similarly, the mutation G208A was reported in four isolates but was synonymous and not expressed into proteins. Hence, our study indicates that three nucleotide positions, viz., 211, 480, 491 and 208, are critical in the Pakistani scenario.

The application of genome analysis by next-generation sequencing will improve the clinical management of TB, especially MDR-TB. This technology will be helpful in the endemic regions of TB and also best for disease surveillance. These rapid and highly reliable methods identify critical and known mutation markers. This advanced technology will also be helpful to discover mutation in mixed infections, hetero resistance, and transmission setting. This advanced technology is also used to assist and understand the epidemiology and factors involved in the transmission of TB. It also provides valuable information about mutations in MTBC against different drugs. The regimen for TB is also offered on the basis of mutation and early diagnosis and knowing of resistance patterns of MTBC, which will be helpful in disease control strategies in high-burden countries, especially in Pakistan. The limitation of this study was that we did not perform vet lab for secretory proteins.

## 5. Conclusions

MPT64 is highly conserved secretory protein, harboring only few mutations. This study provides useful information for the better understanding about virulency and diagnosis of MTBC isolates in high TB burden countries. MPT64 protein can be considered to be a potent vaccine candidate.

## Figures and Tables

**Figure 1 ijerph-20-02530-f001:**
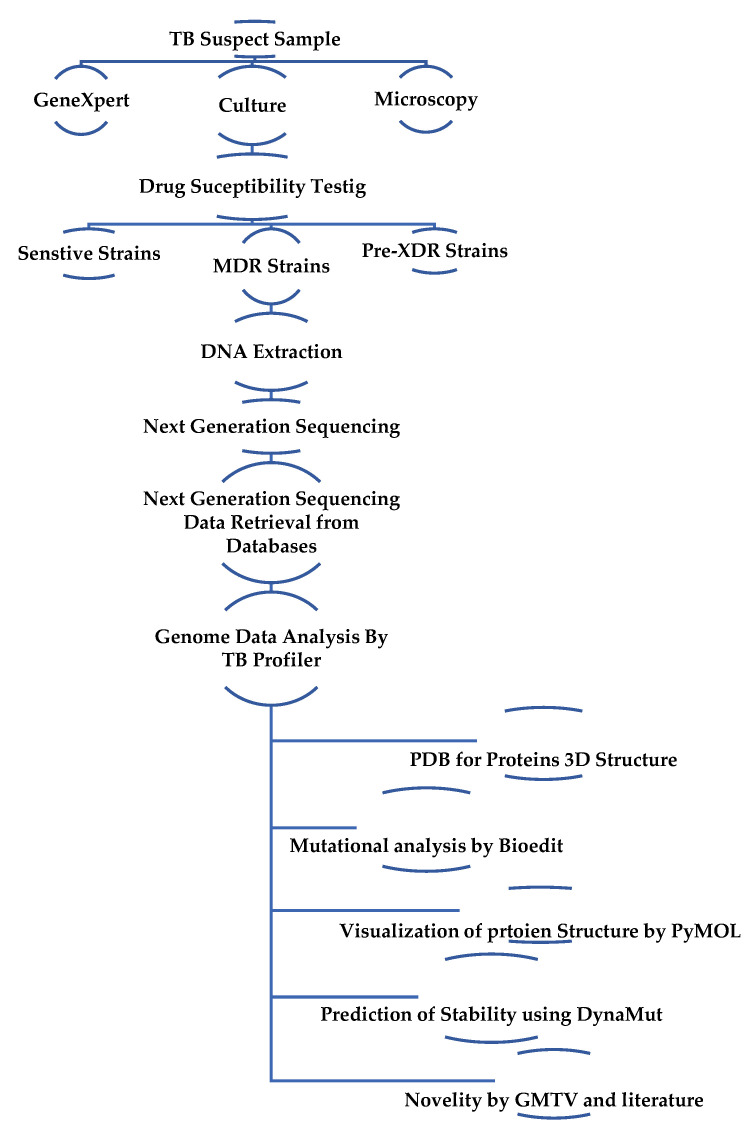
Methodology flowchart.

**Figure 2 ijerph-20-02530-f002:**
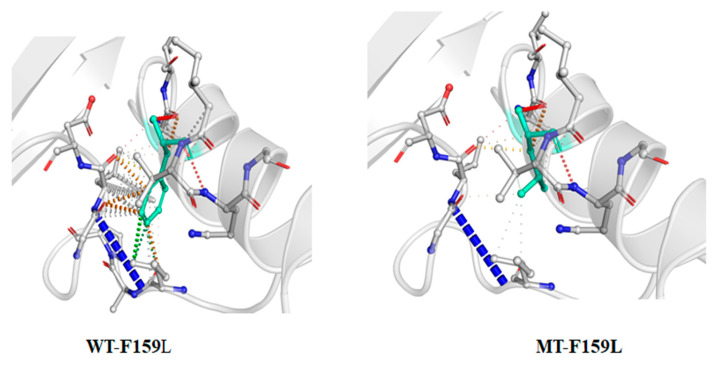
Wild and mutant type structure of F159L. The light blue residue is a mutation site. WT—wild type; MT—mutant. WT exhibited more interaction at F159, whereas MT L159 showed fewer interactions.

**Figure 3 ijerph-20-02530-f003:**
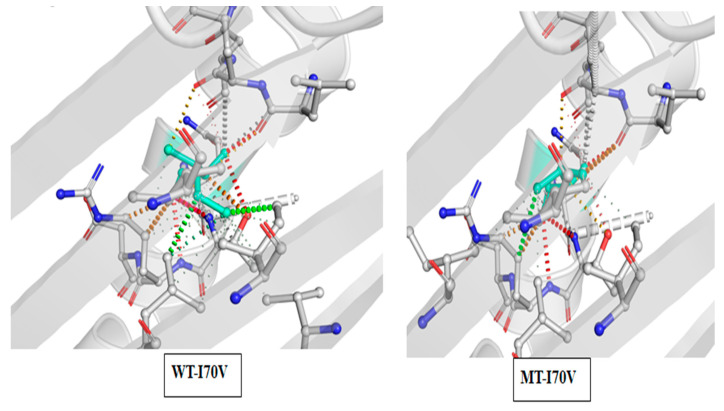
Comparison of WT and MT I70V interactions at mutation site residues. The interaction frequency of WT I70 exhibited numerous bonds with surrounding residues. The MT has fewer interactions at V70 (light blue) than WT.

**Figure 4 ijerph-20-02530-f004:**
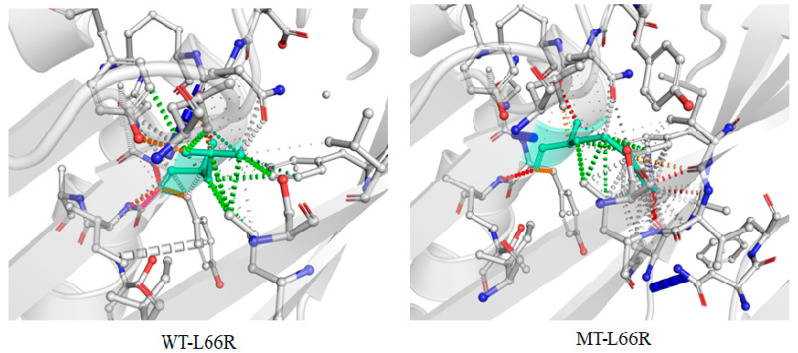
Wild and mutant type structure of L66R. The interaction frequency of WT is fewer than (light blue) than that of MT.

**Table 1 ijerph-20-02530-t001:** Mutations in MPT64 protein.

Position of Nucleotide/Amino Acid	Frequency	Accession No.
G211T (F159L)	6	ERR3335730, ERR3335746, ERR2510249, ERR2510255, ERR2510448, ERR2510512
T480C (I70V)	2	ERR2510511, ERR2510860
A491C (L66R)	1	ERR3335711
G208A	4	ERR3335, ERR2510553, ERR2510554, and ERR2510557

**Table 2 ijerph-20-02530-t002:** Drug susceptibility pattern of non-synonymous mutated isolates.

S. No.	Age	Gender	Accession No.	Mutation	Treatment History	S	I	R	E	P	O	M	A	K
1	32	M	ERR3335730	F159L	Cate-I	S	S	S	S	S	R	S	S	S
2	15	M	ERR3335746	F159L	Cate-I	S	S	S	S	S	S	S	S	S
3	25	F	ERR2510249	F159L	Cate-II	S	R	S	S	S	S	S	S	S
4	26	M	ERR2510255	F159L	Cate-II	R	R	R	R	R	NA	NA	NA	NA
5	60	M	ERR2510448	F159L	Cate-II	S	R	R	R	R	NA	NA	NA	NA
6	23	M	ERR2510512	F159L	Cate-I	S	R	R	R	R	NA	NA	NA	NA
7	10	F	ERR2510511	I70V	Cate-I	S	R	R	R	R	NA	NA	NA	NA
8	38	F	ERR2510860	I70V	Cate-I	S	S	S	S	S	NA	NA	NA	NA
9	46	M	ERR3335711	L66R	Cate-I	S	R	R	S	S	R	R	S	S

Note: S—Streptomycin; I—Isoniazid; R—Rifampicin, E—Ethambutol; P—Pyrazinamide; O—Ofloxacin; M—Moxifloxacin; A—Amikacin; K—Kanamycin.

**Table 3 ijerph-20-02530-t003:** Detailed illustration of positive isolates and mutations in MPT64 gene.

Sample No	Culture	Smear	ICT	Line Probe Assay	Position of Mutation
N1	+	+	+	+	G211T
N2	+	+	+	+	G211T
N3	+	+	+	+	G211T
N4	+	+	+	+	G211T
N5	+	+	+	+	G211T
N6	+	+	+	+	G211T
N7	+	+	+	+	T480C
N8	+	+	+	+	T480C
N9	+	+	+	+	A491C
N10 (Control)	+	+	+	+	No mutation

**Table 4 ijerph-20-02530-t004:** Effect of mutation on stability and flexibility of the proteins.

Mutation	No of Isolates	Prediction Outcome ΔΔG kcal/mol	Effect on Protein Structure	Entropy Energy between * WT and MT (ΔΔSVib (kcal·mol^−1^·K^−1^))	Flexibility
G211T (F159L)	6	−1.855	Destabilizing	0.909	Increase
T480C (I70V)	2	−0.381	Destabilizing	0.23	Increase
A491C (L66R)	1	−0.798	Destabilizing	0.931	Decrease

* WT—wild type; MT—mutant.

## Data Availability

The data sets supporting the results of this article are available in the National Center for Biotechnology Information (NCBI) (https://www.ncbi.nlm.nih.gov/), for the Pakistani isolates with the following accession numbers: ERR5897071-ERR5897121, dated 5 November 2021 of KP isolates and ERR2510337-ERR2510768 dated 10 November 2020, of the project San Raffaele Scientific Institute (CRyPTIC).

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
