# Peer review of "Novel Mutations in MPT64 Secretory Protein of Mycobacterium tuberculosis Complex"

_ijerph, 2023, doi:10.3390/ijerph20032530_

Round 1
Reviewer 1 Report
MPT64 is one of the fastest strips to diagnose the TB. The authors investigated mutations in MPT64 protein which is the main diagnostic biomarkers in tuberculosis patients. The study used whole genome sequences for mutations in MPT64. The authors also investigated the stability effect along with dynamics on proteins. Some novel mutations have been reported. MPT64 is a diagnostics marker and mutations screening and reporting may be useful to assess the diagnostic process.
The study is very important for better management of TB. However, discussion can be improved for reader interest with addition of some relevant references. Some mutational data in MPT64 may be added and discussed of found in previous studies. A flow chart methodology may be added to better understanding. Some minor grammatical errors may be removed for clear look.
Author Response
Response to reviewer 1 comments

Reviewer 2 Report
Dear Authors,
This manuscript will contribute to science but it needs some corrections. You can view these revisions in the attached PDF file. It can be published after corrections.
Best regards

Author Response
Response to reviewer 2 comments

Reviewer 3 Report
Comments
In this research article, the authors aimed to find a mutation in this highly conserved protein in isolates from the Pashtun-dominant province of Pakistan. TB-Profiler and BioEdit software tools were used to screen mutations in MPT64 gene, DynaMut web server was used to analyze the impact of the mutation on protein dynamics and stability. The manuscript is unique and original, experimental design and data analysis are robust. I recommend the following comments to the authors.
Major points
1. The author must clarify the numbers of retrieved and processed samples, as you mentioned 470 retrieved samples in the abstract and methodology.
2. Updated definitions and treatment of XDR TB as per the WHO guidelines are missing.
3. The author should explain the results of capilia and which type of capilia was used for diagnosing MTB. Unfortunately, the introduction and the discussion lack information about the capilia used.
Minor points
1. Rephrase/rewrite the sub-title 2.5 and mention the effect of the mutation.
2. The author must arrange table no 2 in results, in a sequence like→S.No→Age of patients→gender→ accession no and mutation.
3. The author has to write the full names of anti-TB drugs in results table 2.
4. The author must also explain the WT and MT in Fig: 1.
5. In discussion paragraph no 2, the ICT should be written as immunochromatographic technique.
Author Response
Response to reviewer 3 comments

Reviewer 4 Report
The authors investigated MPT64 mutation using M. tuberculosis whole genome sequences and computed mutation effect on structural stability.
The authors report novel mutations in MPT64 in a large number of number genomic isolates. Such types of study is always interesting for the readers, as well as MPT64 conservations, is checked to observe its diagnostics accuracy.
These types of studies should be performed for accurate diagnostics, especially in high-burden countries. This study is interesting however, some minor errors may be removed, and some parts could be improved before its publication. The citations should be revised.
TB abbreviation should be defined when it is used for the first time in the introduction.
Some description of the project San Raffaele Scientific Institute (CRyPTIC) should be added Methodology.
Section 2.4. The crystal structure of MPT64 has been retrieved from the protein data bank under the accession code 2HHI? A little description if PDB should be added.
Some minor grammatical check is required in the result section.
The discussion section could be improved with the addition of some data. The MPT64 description and role in diagnosis should be elaborated here. This section needs some previous mutational data and description of those studies to be added. Stability may also be discussed for reader interest.
Author Response
Response to reviewer 4 comments
